# Stakeholders’ Perceptions of Biosecurity Implementation in Italian Poultry Farms

**DOI:** 10.3390/ani13203246

**Published:** 2023-10-18

**Authors:** Andrea Laconi, Giuditta Tilli, Francesco Galuppo, Guido Grilli, Rozenn Souillard, Alessandra Piccirillo

**Affiliations:** 1Department of Comparative Biomedicine and Food Science, University of Padua, Viale dell’Università 16, 35020 Legnaro, Italy; andrea.laconi@unipd.it (A.L.); giuditta.tilli@phd.unipd.it (G.T.); 2Unità Locale Socio-Sanitaria (ULSS) 6 Euganea, Via Enrico degli Scrovegni 14, 35131 Padua, Italy; francesco.galuppo@aulss6.veneto.it; 3Department of Veterinary Medicine and Animal Science, University of Milan, Via dell’Università 6, 26900 Lodi, Italy; guido.grilli@unimi.it; 4Epidemiology, Health and Welfare Unit, French Agency for Food, Environmental and Occupational Health & Safety, 22440 Ploufragan, France; rozenn.souillard@anses.fr

**Keywords:** poultry, farmers, advisors, biosecurity, perception

## Abstract

**Simple Summary:**

In poultry production, biosecurity is essential for preventing the entry and spread of pathogens in/within farms, with positive impacts on animal health and welfare. This becomes crucial in countries in which the production reaches high productive standards, such as Italy. Although the importance of implementing biosecurity measures (BMs) in poultry farms is well recognized, compliance may not be always optimal. According to the perceptions of interviewed Italian poultry farmers and advisors, biosecurity is implemented well, with different opinions between them. Compared to farmers, advisors claim a lower level of biosecurity implementation, probably because the former have limited outlooks focused on their own farms or consider some BMs not useful and/or effective, while advisors possess a more comprehensive picture of the implementation of biosecurity due to their involvement in multiple farms and productive categories. Stakeholders, mainly advisors, provided useful suggestions for encouraging farmers and workers at poultry farms to adopt BMs they remain reluctant to implement.

**Abstract:**

The level of implementation of biosecurity measures (BMs), the reasons for not implementing BMs and the effectiveness of BMs were assessed according to the perceptions of stakeholders (i.e., farmers and advisors) in Italian poultry farms. For this purpose, data were collected using a questionnaire administered to advisors (*n* = 37) and farmers (*n* = 30) of conventional broiler (*n* = 13) and layer (*n* = 13), free-range broiler (*n* = 8) and layer (*n* = 10), turkey (*n* = 13), duck (*n* = 3) and breeder (*n* = 7) farms between April and September 2021. The frequency of the implementation of BMs was 66.97% and 81.14% according to the answers provided by the advisors and farmers, respectively, with the breeder sector showing the highest level of implementation (85.71%). “Not knowing advantages” (21.49% for advisors) and “other/specific reasons” (21.49% for advisors and 38.32% for farmers) were the most common answers regarding the lack of implementation of BMs for all poultry sectors. Only 31.09% of farmers acknowledged the effectiveness of not-implemented BMs in contrast to 61.02% of advisors, with the layers’ stakeholders being the most aware. The findings of this study may be useful for identifying failures in biosecurity and failures to develop intervention strategies to fulfil the biosecurity gaps still present in Italian poultry farms.

## 1. Introduction

Italy is the fifth and the fourth largest producer of poultry meat and eggs in Europe (EU), with approximately 1.21 M and 744 M tons of meat and eggs produced in 2022, respectively (https://www.unaitalia.com/, accessed on 1 June 2023). Italian poultry production is organized in vertically integrated companies, with most poultry (e.g., chickens, turkeys and layers) farms concentrated in the northern regions of the country (i.e., Veneto, Emilia Romagna and Lombardia). In commercial poultry farms, mortality rates due to infectious diseases, such as Avian Influenza (AI) [1], can be high, while the contamination of poultry meat and eggs by zoonotic agents, such as *Salmonella* and *Campylobacter* [2,3], can be harmful for public health. Therefore, the prevention and control of infectious agents in the poultry sector are essential to reducing animal and economic losses at the farm level and preserving food safety for consumers.

Disease prevention through the implementation of proper biosecurity measures (BMs) in livestock production may have beneficial effects not only for animal health but also for human and environmental health, which can be summarized in the concept of “One Biosecurity” [4]. The prevention of infectious diseases can be achieved by adopting several different interventions (e.g., management and vaccination), and biosecurity represents one of the most effective ways to manage the risks posed by the introduction and spread of infectious agents [4]. Although many definitions of biosecurity have been provided so far, it is now considered a holistic and integrated approach involving different stakeholders and sectors [5]. Biosecurity is a complex strategy requiring the implementation of several different practices at any level of livestock farming [6]. In the poultry sector, biosecurity is applied along the entire production chain, from breeders to the slaughterhouse [7,8]. Since each step of the production chain is strictly linked to the following one, the risk of the transmission of infectious agents can be easily encountered, and compliance with biosecurity should be ensured at each step [9]. BMs can be classified into two main categories: external biosecurity aimed at preventing the introduction of pathogens into a farm (e.g., farm delimitation, the entry of people and vehicles, feed and drinking water management) and internal biosecurity aimed at reducing the spread of pathogens within a farm (e.g., hygiene lock, cleaning and disinfection and sanitary breaks) [10,11]. 

To minimize the risk of the entry and spread of infectious agents in/within farms, biosecurity should rely on different components at conceptual, structural and operational (or procedural) levels [12]. While the conceptual and structural components of biosecurity are well established at the time of farm and production design, procedural biosecurity requires the constant implementation of routine practices, envisaging an active role of the farmer and all the workers involved in production [12,13]. In order to be fully compliant, the implementation of biosecurity should constantly cover all three levels, with the procedural level being the most difficult since it relies on the human factor [14]. Indeed, the factors affecting compliance with procedural BMs include the attitudes and behaviors of people working at and/or visiting the farms, which can challenge the final goal of biosecurity, i.e., preventing pathogens from harming the health and the welfare of poultry flocks [13]. Since human behavior and attitude are thought to be essential for the implementation of biosecurity, understanding the perceptions of the stakeholders of the poultry sector toward BMs and their effectiveness is of the utmost importance, as previously described in several studies [7,13,15]. 

Given the above considerations, the aims of this survey were to (a) assess the implementation of BMs in Italian poultry farms according to farmers’ and advisors’ perceptions; (b) investigate the reasons for which BMs are not implemented; and (c) identify whether not implemented BMs are perceived as effective.

## 2. Materials and Methods

### 2.1. Target Population and Data Collection

Data were collected between April and September 2021 via questionnaires which were developed within the European NetPoulSafe project (G.A. 101000728) and administered to 67 randomly selected farmers and advisors from different sectors, i.e., poultry companies’ veterinarians (*n* = 12), local public institutions (*n* = 6), academic experts (*n* = 5), other poultry veterinarians (*n* = 6), and producers’ organizations (*n* = 8). The selected farms were located in the most relevant areas for national poultry production, i.e., Veneto (*n* = 19), Lombardia (*n* = 5), Emilia Romagna (*n* = 3), Piemonte (*n* = 2) and Puglia (*n* = 1). The interviewed advisors provided answers to the questionnaire based on their constant interactions with many poultry farms (for some advisors >1000 farms) which are distributed all over the Italian territory. The questionnaires were completed during physical or video call interviews. In detail, the questionnaires were administered to *n* = 30 farmers and *n* = 37 advisors of conventional broiler (*n* = 13) and layer (*n* = 13), free-range broiler (*n* = 8) and layer (*n* = 10), turkey (*n* = 13), duck (*n* = 3) and breeder (*n* = 7) farms (Table 1), all belonging to Italian integrated companies. 

The questionnaire included a set of 38 questions on BMs common to all poultry sectors, with additional questions specific to the various productive categories (*n* = 9, free-range; *n* = 8, breeders; *n* = 7, layers) which were included in separate sections of the questionnaire. The common section was composed of 8 items including 26 external BMs and 4 items including 12 internal BMs. The questions and answer options for each BM present in the questionnaire are represented in Figure 1. The questionnaire administered to farmers (Appendix A) was slightly different from the questionnaire administered to advisors (Appendix A). In this paper, only answers to the common section of the questionnaire are reported and discussed.

### 2.2. Data Analysis

The answers obtained from the common section of the questionnaire were collected in a database (Sphynx^®^ software, https://www.lesphinx-developpement.fr/, accessed on 30 January 2023), extracted in Excel^®^ and analyzed using descriptive statistics to assess the farmers’ and advisors’ perceptions of the level of implementation of BMs, the reasons for not implementing BMs and the efficacy of the BMs that were not implemented. A cut-off value of 70% [16] was used to classify BMs as highly (>70%) or scarcely (≤70%) implemented according to the cumulative score of the answers provided by the farmers and the advisors. Chi-squared and Fishers’ exact tests were used to assess the significance of the bivariate relationship between the different poultry sectors (i.e., breeders, broilers, free-range broilers, layers, free-range layers and turkeys) and the different stakeholders (i.e., farmers and advisors), respectively. Due to the low number of observations, duck farms were excluded from the bivariate analysis. Differences were considered significant if the *p*-value was <0.05. The statistical analyses were performed using GraphPad Prims v10.0.1 software (http://www.graphpad.com, accessed on 30 June 2023). 

## 3. Results

### 3.1. Level of Implementation of BMs as Perceived by Farmers and Advisors

According to the “always” answers provided by the interviewed stakeholders, an overall high level of biosecurity implementation (74.99%, 95% confidence of interval (CI) 73.05–76.83%) was detected; however, a significant (*p* < 0.0001) difference between the farmers’ (81.14%, 95% CI 78.77–83.31%) and advisors’ (66.97%, 95% CI 63.79–70.01%) responses was noticed (Figure 2A). According to the interviewed stakeholders, internal BMs (82.73%, 95% CI 78.48–86.29% and 69.15%, 95% CI 63.39–74.36%, respectively) were more implemented than external ones (80.51%, 95% CI 77.59–83.14% and 64.10%, 95% CI 60.26–67.77, respectively) (Appendix A).

Among the different poultry sectors, the interviewed stakeholders stated that BMs were implemented as follows (from the highest to the lowest): breeders (85.71%), ducks (81.58%), turkeys (76.42%), broilers (72.82%), layers (72.45%), free-range broilers (71.95%) and free-range layers (69.84%) (Table 2 and Figure 2B). Significant (*p* < 0.0001) differences were found among all the different poultry productive categories. Between farmers and advisors, significant differences were detected in broilers (*p* = 0.0049), layers (*p* < 0.0001) and free-range layers (*p* < 0.0001) (Figure 2C). 

Out of 38 BMs, the interviewees’ answers revealed that 23 and 15 were highly and scarcely implemented, respectively, with statistically significant differences between farmers and advisors in four highly and eight scarcely implemented BMs (Table 2 and Appendix A). Eleven external and four internal BMs were found to be highly implemented in all poultry sectors, whereas two external BMs and one internal BM were scarcely implemented (Figure 3 and Table 2).

Differences among the poultry sectors were also noticed. In detail, six BMs were scarcely implemented in all sectors except one for breeders and one for turkeys. In contrast, three external BMs were scarcely implemented only in free-range layers, and one internal BM was scarcely implemented only in broilers. It is of note that most of the remaining nine BMs were scarcely implemented, mainly in layers and/or free-range layers (Figure 3 and Table 2). 

### 3.2. Reasons for Not Implementing BMs According to the Farmers’ and Advisors’ Perceptions

When stakeholders did not reply “always” to Q1 (Figure 1), the questionnaire continued by asking for the reasons for which the BM was not implemented. Overall, most of the answers fitted with the options suggested by the questionnaire, even though 25.90% (95% CI 22.65–29.44%) of the stakeholders provided “other/specific reasons” as a reason for not implementing BMs (Figure 4A and Appendix A). 

The farmers were more likely to favor the option “other/specific reasons” compared to the advisors (38.32% 95% CI 31.29–45.88% vs. 21.49, 95% CI 18.01–25.43, *p* < 0.0001), as well as the “not useful” answer (29.94% 95% CI 23.51–37.27% vs. 7.23% 95% CI 5.22–9.94%, *p* < 0.0001). In contrast, the advisors suggested the following answers as reasons for not implementing BMs more frequently than farmers: “take too much time” (16.38% 95% CI 13.31–19.99% vs. 2.99%, 95% CI 1.29–6.81, *p* < 0.0001), “too expensive” (12.98% 95% CI 10.24–16.32% vs. 2.99%, 95% CI 1.29–6.81, *p* < 0.0001) and “not knowing the advantages” (21.49% 95% CI 18.01–25.43% vs. 7.19%, 95% CI 4.16–12.14, *p* = 0.0001).

Considering the different poultry sectors, the answers provided by the interviewed stakeholders confirmed that “other/specific reasons” was the most common option among breeders (26.83%, 95% CI 15.69–42.93%), broilers (27.27%, 95% CI 19.47–36.77%), free-range broilers (35.94%, 95% CI 25.29–48.18%), layers (19.83%, 95% CI 13.59–28%), free-range layers (21.74%, 95% CI 15.18–30.12%) and turkeys (35.29%, 95% CI 25–47.16%) (Figure 4B). The option “other/specific reasons”, together with “take too much time” and “not knowing the advantages”, showed similar frequencies among the different poultry categories, whereas “not adapted to the farm” and “not enough trained” were not indicated by the turkey and breeder farms’ stakeholders, respectively. Additionally, “not useful” was an uncommon answer provided by the layer and free-range layer farms’ stakeholders (*p* = 0.0004). The frequency of the answers provided by farmers and advisors within the same productive category (Figure 4C) was comparable to that reported in Figure 4A. Indeed, the most common answer provided by the farmers in all the productive categories was “other/specific reason”, while the advisors provided the following answers: “take too much time”, “too expensive” and “not knowing the advantages”. 

### 3.3. Opinion of Farmers and Advisors on the Effectiveness of BMs Not Implemented in Poultry Farms

The stakeholders’ perceptions of the efficacy of BMs which were not implemented was requested when they did not reply “always” to Q1 (Figure 1), and more than half (51.47%, 95% CI 46.41–56.51%) of the farmers and advisors recognized their effectiveness (Figure 5A). However, the advisors were more likely to appreciate the beneficial effects of unimplemented BMs than the farmers (*p* < 0.0001). Among the different productive categories, the layer and free-range layer farms’ stakeholders acknowledged the importance of implementing BMs more commonly than the other poultry sectors’ stakeholders (*p* = 0.0456) except for breeder farmers, who declared an “other opinion”.

## 4. Discussion

In poultry production, interventions to prevent and/or control the introduction and spread of pathogens, such as AI viruses and *Salmonella*, rely mainly on the proper implementation of BMs [3,17]. Biosecurity comprises different components, some of which are obviously factual, while others depend on practices routinely applied by people working on the farms [13,18]. In Italy, these elements of biosecurity are usually assessed via farm inspections carried out by official veterinarians who interview farmers and workers and are reported in specific checklists [16]. In the present study, a different questionnaire was used to assess the level of implementation of BMs in Italian poultry farms and the importance of each BM in achieving biosecurity compliance. In order to capture different perceptions, two different stakeholders, namely, farmers and advisors, were interviewed, considering that the former might have focused and/or limited visions of their own situations, while the latter were more likely to have broader and more comprehensive views of the BMs implemented in poultry farms.

In a previous study [16], we found that biosecurity compliance in broiler, layer and turkey farms located in a densely populated poultry area of northeastern Italy was high, with external BMs more implemented than internal ones. In the current study, the interviewed stakeholders confirmed high levels of biosecurity implementation in the same poultry categories, as well as in additional ones (i.e., free-range broiler and layer, duck, and breeder farms). Indeed, out of the 38 BMs investigated, 23 (15 external and 8 internal) were declared to be highly implemented, and this might be attributed to the periodic and systematic inspections of poultry farms carried out by the integrated companies and the official veterinarians according to the national legislation, which has been in force since 2005 [19]. Contrary to our previous observations [16] but in agreement with other European studies [11,20], in the present study, the frequency of the implementation of internal BMs was stated to be slightly higher than the implementation of external BMs (76.91% and 73.22% of highly implemented BMs, respectively) by the interviewed stakeholders. The contrasting findings observed in the two Italian studies are probably due either to the different questionnaires (official checklist vs. ad hoc questionnaire) used and/or the different geographical areas (provincial vs. national area) considered. The highest levels of biosecurity implementation were detected in breeder farms, according to both farmers’ and advisors’ opinions. Indeed, six BMs (i.e., personnel, visitors and/or teams washing hands and showering before entering the house, no domestic animals on site and concrete surrounds around the house) were scarcely implemented in any of the production categories except for breeders. These findings were not unexpected since breeders are at the top of the production pyramid and therefore, more attention is given to their management compared to other poultry categories. Indeed, given the high economic value and the risk to vertically transmit important pathogens, such as *Salmonella* and *Mycoplasma*, these birds are usually reared under strict biosecurity conditions [21,22]. Notably, the breeder farms’ stakeholders never declared “not enough trained” as a reason for not implementing BMs, confirming the high level of attention to fulfilling biosecurity requirements.

Among all the different poultry sectors, the farmers and advisors agreed on a high level of implementation of BMs in most of the procedures related to the movement of people (i.e., personnel, visitors and teams) on the production site; the presence of biological vectors (i.e., backyard animals, rodents and wild birds); cleaning and disinfection (i.e., cleaning the vehicle wheels, house, material, feed silo and the drinking water pipeline between cycles); flock traceability (i.e., registering flocks) and health surveillance (i.e., vaccination and clinical signs and mortality monitoring); the storage of feed; and dead animals. On the other hand, some scarcely implemented BMs regarding the circulation of people (i.e., using specific clothes and shoes for the chick deliverer) on the production site and conducting a microbiological screening following cleaning and disinfection procedures for the poultry house were also acknowledged by the two interviewed stakeholders of all poultry sectors. Overall, these findings suggest that the interviewed poultry stakeholders are aware of the importance of implementing both external and internal biosecurity measures, according to our previous findings [16]. In detail, BMs relating to the circulation of people, vehicles and equipment, as well as biological vectors, were revealed to be almost always implemented. This is a valuable achievement since they represent significant risk factors for the introduction and spread of pathogens in/within poultry flocks [17,23,24]. Even though most of the cleaning and disinfection procedures were declared to be highly implemented, the same did not apply to the microbiological analysis of the poultry house carried out at the end of the production cycle in order to assess their effectiveness. To note, most of the stakeholders attributed (as answered in “other/specific reasons”) the poor implementation of this measure to the integrated companies that usually use validated disinfection protocols. This practice, however, should be encouraged given the importance of verifying both the actual implementation and the efficacy of the procedure [11], and this was recognized by many of the interviewed stakeholders. Feed and water management was overall indicated as compliant; however, some gaps in its correct implementation (e.g., conducting a drinking water analysis) were reported by the interviewed stakeholders. Notably, some stakeholders specified that water analyses were performed with “other frequencies” (i.e., twice a year), while others believed it to be “not useful” because the water source was the aqueduct. Analysing water quality is a fundamental aspect of avoiding any contamination of poultry flocks, not only from pathogens but also from chemical contaminants, such as antimicrobials and their related resistances [1], and the interviewed advisors were fully aware of its importance. For many animals (e.g., wild birds, rodents and insects) that can be vectors of poultry pathogens [3,25,26], the attention of the people working on the farms seems to be high, except for domestic animals (e.g., dogs and cats). The proximity of the farmer/worker house to the poultry site is the main reason why domestic animals have access to the farm, as detailed by the advisors in “other/specific reasons” (together with “not knowing the advantages”). However, this faulty habit should be discouraged [27], as was also suggested by the interviewed advisors. Many of these BMs have been demonstrated to be effective in preventing the dissemination of poultry pathogens, such as AI viruses [17,28], *Salmonella* [3,29] and *Campylobacter* [30,31] on a poultry site. Therefore, interventions such as cleaning and disinfection and restricting access to the farm/house are deemed fundamental to tackling this challenges and preserving the flock’s health [32].

Some interesting differences in the implementation of BMs were declared by the stakeholders of the various poultry sectors, with broiler and layer farms, including the free-range ones, showing lower levels of implementation of specific external and/or internal BMs when compared to the other poultry categories. In some cases, the non-compliance of some BMs could be attributed to the specific requirements of a given production category, rearing system or other reasons rather than the unwillingness of the farmers. For example, the “all-in/all-out” system was stated to be fully implemented in all poultry sectors except for the enclosed and free-range layer farms (i.e., multi-age), in which this practice is not commonly applied for commercial reasons. However, the interviewed stakeholders pointed out that in Italy, the all-in/all-out system is mandatory by law on layer farms (as detailed in “other/specific reasons”), even though they recognized the great importance of “all-in/all-out” system in avoiding the transmission of pathogens among different ages and houses. In addition, the non-compliance of broiler farms with a sanitary break longer than 15 days can be explained by the compulsory sanitary break of 7 days foreseen by the Italian legislation in this productive category. Similarly, the cleaning and disinfection of the feed silo at the end of each cycle is believed to not be useful because of the broilers’ short cycle. Some deficiencies in the implementation of BMs were detected only on free-range broiler and layer farms (e.g., the access of people and vehicles and the management of dead animals). Even if the implementation of some of these BMs might not be straightforward in this farming system, according to the farmers’ opinions (e.g., personnel changing clothes in the outdoor farming area), others could be easily implemented (e.g., the management of dead animals) and should be encouraged since they are crucial to reducing the risk of transmitting disease in free-range poultry [12]. However, the low levels of compliance with some BMs (e.g., the protection of litter and the location of the rendering tank, manure, etc., outside of the clean area) may be due to the lack of usefulness of the practice itself (e.g., the lack of a need to store these elements) or to the structural characteristics of a poultry site (especially aged farms). In this case, the stakeholders stressed the need for financial support to renew structural deficiencies in biosecurity compliance.

The questionnaire used in this study was designed and structured to be answered by stakeholders at the European level rather than at the Italian level. For this reason, some questions did not consider the peculiarities of the Italian poultry farming system and the national legislation. For example, “if other animal productions on the site (cattle, pigs) sanitary barriers with poultry (personal, material …)” was declared to be scarcely implemented by all the poultry farmers and advisors. On the other hand, it should be empirically considered to be highly implemented since in Italy, the presence of different types of animal production on the same site has been historically strongly discouraged by both official veterinary services and integrated companies, as stated by the interviewed stakeholders. Similarly, the unavailability of specific clothes and shoes for chick deliverers should not be considered a breach of biosecurity measures because these workers are usually not allowed to enter the farm, as also detailed in “other/specific reasons” by the interviewed stakeholders. Furthermore, individual practices related to the entry of people to the poultry site yielded contrasting answers because some were declared to be highly implemented (i.e., registering visitors and teams; the use of specific shoes and clothes), while others were stated to be scarcely implemented (i.e., washing hands and showering before entering the house). Indeed, these procedures were considered “not useful” by the farmers and “time-consuming” and “not knowing advantages” by the advisors. In Italy, the individual BMs for entering a farm and house are regulated by national legislation, according to which some BMs are mandatory while others not, with access to the farm (i.e., filter zone) more strictly regulated than entry to the house (i.e., the Danish entry system). While the farmers consider the house hygiene lock “not useful”, the advisors stated that it is “time-consuming” and “too expensive”. Overall, both groups of stakeholders were in agreement on the effectiveness of the Danish entry system. Moreover, Italian law does not require the presence of a hygiene lock with two separate zones at the entrance of each house, and this might explain why this BM was declared to be scarcely implemented. Moreover, even though the farmers recognized the importance of keeping the surroundings of the house clean, they declared that cutting the grass, removing the gravel and avoiding the presence of waste materials are as effective as using concrete floor, according to the Italian legislation, which requires only a concrete area in front of the house. On the other hand, a concrete area surrounding the poultry house would be “too much expensive” for some poultry categories (e.g., broilers and layers), as per the advisors’ opinions.

In this study, a significant difference between the farmers’ and advisors’ perceptions of biosecurity was detected. Indeed, the farmers and advisors did not agree (*p* < 0.05) on the levels of compliance of both highly (e.g., the movement of vehicles and equipment on the production site) and scarcely (e.g., the presence of other animal productions on site) implemented BMs. While some of these BMs can be easily verified (e.g., protecting windows with nets and closing and protecting the rendering tank), others (e.g., cleaning and disinfection procedures and the use of clothes and shoes dedicated to each house) are less verifiable because they rely on statements given by the farmers or workers, and this should be always taken into account when assessing biosecurity compliance, as previously described in several papers [7,13,15,18]. Indeed, the farmers were more inclined to state that the farms were compliant with biosecurity standards than the advisors. Furthermore, the farmers may have views limited to their own farms, unlike the advisors. Notably, in Italy, several BMs are mandatory by law (e.g., the use of specific shoes for personnel and visitors), and this might have influenced the reliability of the answers provided by the farmers. On the other hand, some personal hygiene procedures, such as hand washing and showering, that fall into procedural biosecurity, were declared to be scarcely implemented by the stakeholders of all the production categories except for breeders. However, the same stakeholders recognized the positive impact of these practices on reducing the risk of introducing pathogens into flocks. Previously [16], we speculated that farmers’ answers might sometimes be arguable when answering to official audits. In this study, however, the use of a different questionnaire administered during unofficial interviews and supported by the advisors’ opinions provided comparable results. Even though the interviewed stakeholders were not always in agreement, the consistency of our findings seems to suggest that farmers’ and advisors’ perceptions can be considered trustworthy overall.

In conclusion, the answers provided by the interviewed stakeholders to explain the lack of implementation of certain BMs suggest that advisors are more aware of the importance of achieving high levels of biosecurity compliance, as well as appreciating the effectiveness of BMs compared to farmers. Indeed, the advisors provided useful suggestions for further improving biosecurity compliance in poultry farms (e.g., conducing a microbiological screening after cleaning and disinfection procedures). The farmers commonly defined the unimplemented BMs as “not useful” or “not adapted to the farm”, while the advisors pointed out that the lack of implementation may in fact rely on the unwillingness of the farmers to apply some BMs (i.e., “take too much time”) or their poor knowledge about the benefits of properly implementing BMs (i.e., “not knowing advantages” and “too expensive”). Interestingly, the stakeholders of layer and free-range layer farms declared the lowest levels of BM compliance. However, they appeared to have higher levels of awareness since they rarely chose the option “not useful” (*p* = 0.0004) to justify the lack of implementation of BMs and, at the same time, they were more likely to acknowledge the importance of implementing BMs compared to the other poultry sectors, possibly suggesting their willingness to improve biosecurity.

## 5. Conclusions

Thanks to this study, together with the understanding of farmers’ and advisors’ perceptions of the implementation of BMs, it was possible to identify the reasons for the lack of implementation of some BMs in specific poultry sectors and which measures could be successfully adopted to improve biosecurity compliance. This study was conducted within the NetPoulSafe project; therefore, the selection (i.e., sample randomization) and the number (i.e., a minimum of four and five advisors and farmers, respectively, according to the variety of the national poultry production) of stakeholders to be interviewed per category (a minimum of four categories) were established a priori and agreed upon within the consortium. Even though this might be a limitation of the study, the data collected provide a remarkable insight into the Italian biosecurity status in the context of the European scenario.

An overall high level of biosecurity implementation in all the poultry categories was detected in Italian poultry farms. However, thanks to the answers provided by the interviewed stakeholders, some gaps in biosecurity implementation were identified, suggesting that there is a need to improve the poultry farm workers’ knowledge and awareness. Supporting measures to improve biosecurity compliance (e.g., coaching and group discussions) should be undertaken with the aim of increasing the farmers’ awareness, and therefore willingness, to fully implement biosecurity on their farms.

## Figures and Tables

**Figure 1 animals-13-03246-f001:**
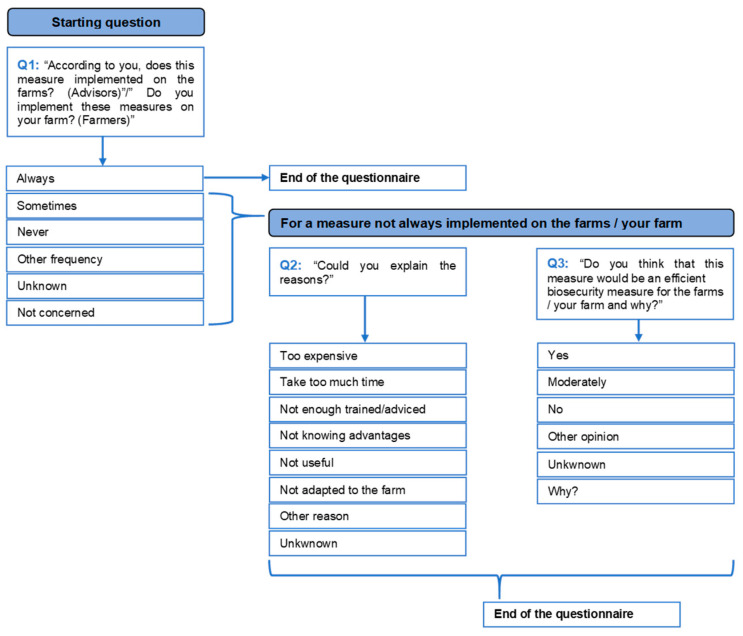
Workflow of the questions and the answer options administered to farmers and advisors through the questionnaire.

**Figure 2 animals-13-03246-f002:**
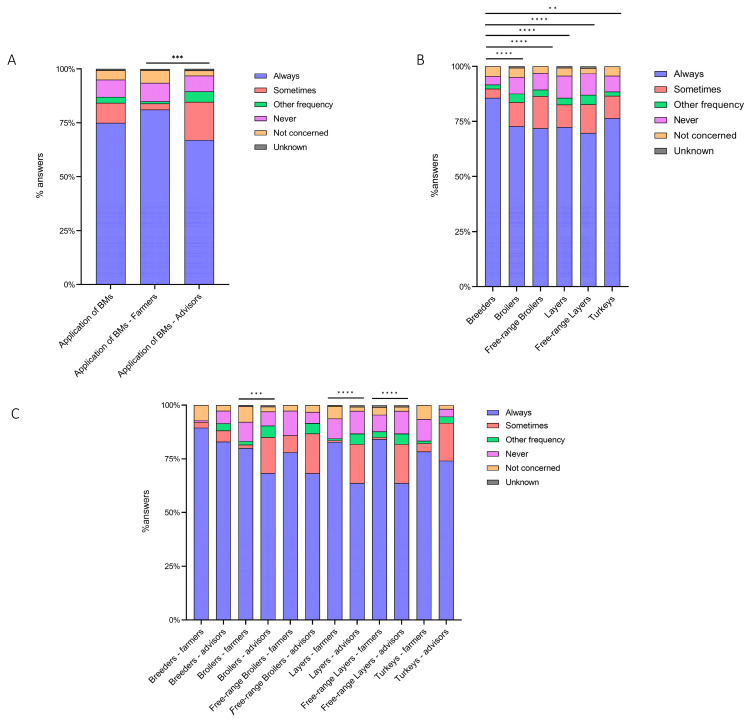
Cumulative percentage of the level of BM implementation, as perceived (**A**) by all stakeholders, farmers and advisors; (**B**) as perceived by all stakeholders in the poultry sector; and (**C**) as perceived by farmers and advisors in the poultry sector. *p* < 0.01 is shown as **, *p* < 0.005 as *** and *p* < 0.0001 as ****.

**Figure 3 animals-13-03246-f003:**
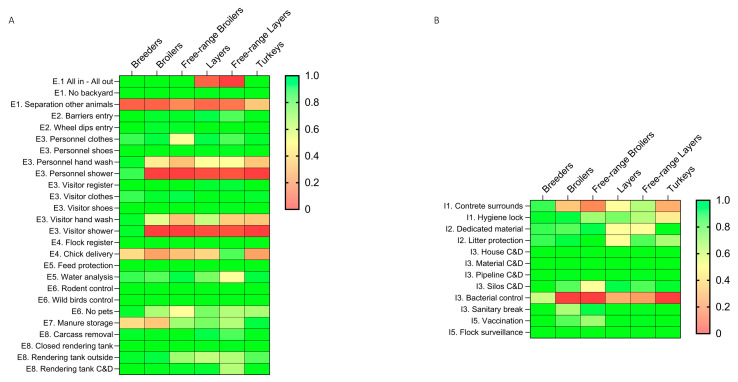
Heatmap showing the level of implementation for each BM, as perceived by the stakeholders according to the poultry sector. (**A**) External BMs; (**B**) internal BMs.

**Figure 4 animals-13-03246-f004:**
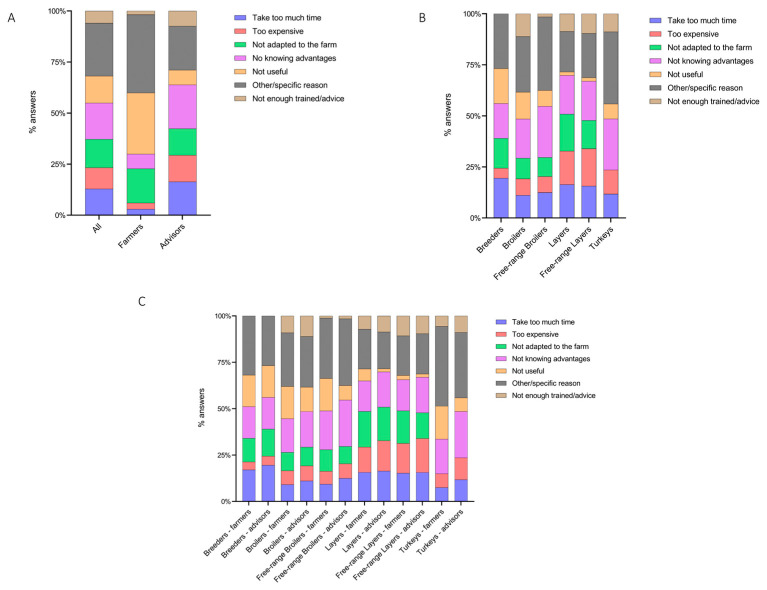
Cumulative percentage of reasons for not implementing BMs, as perceived (**A**) by all stakeholders, farmers and advisors; (**B**) as perceived by all stakeholders according to the poultry sector; and (**C**) as perceived by farmers and advisors according to the poultry sector.

**Figure 5 animals-13-03246-f005:**
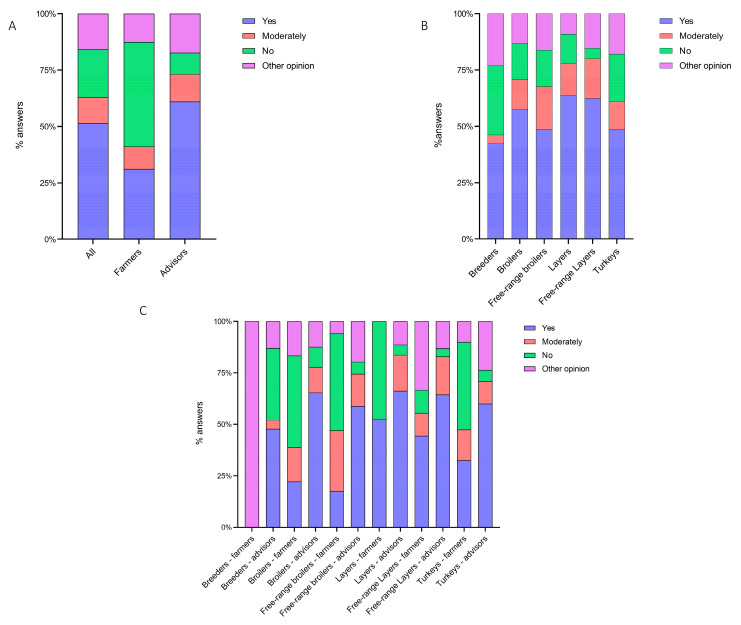
Cumulative percentage of the efficacy of the unimplemented BMs, as perceived (**A**) by all stakeholders, farmers and advisors; (**B**) as perceived by all stakeholders according to the poultry sector; and (**C**) as perceived by farmers and advisors according to the poultry sector.

**Table 1 animals-13-03246-t001:** The demographics and poultry sectors of the interviewed farmers (*n* = 30) and advisors (*n* = 37).

	Farmers	Advisors	Total
Demographics	No. (%)	No. (%)	No. (%)
Gender			
Male	27 (90.0)	30 (81.1)	57 (85.1)
Female	3 (10.0)	7 (18.9)	10 (14.9)
Age group			
<35 years	5 (16.7)	3 (8.1)	8 (11.9)
35–55 years	18 (60.0)	22 (59.5)	40 (59.7)
>55 years	7 (23.3)	12 (32.4)	19 (28.4)
Poultry sectors			
Breeders	3 (10.0)	4 (10.8)	7 (10.4)
Broilers	5 (16.7)	8 (21.6)	13 (19.4)
Free-range broilers	3 (10.0)	5 (13.5)	8 (11.9)
Layers	6 (20.0)	7 (18.9)	13 (19.4)
Free-range layers	3 (10.0)	7 (18.9)	10 (14.9)
Turkeys	7 (23.3)	6 (16.2)	13 (19.4)
Ducks	3 (10.0)	-	3 (4.5)

**Table 2 animals-13-03246-t002:** The frequency of the implementation of external and internal BMs, according to the interviewed poultry farmers and/or advisors (according to the “always” answer).

	Total	Breeders	Broilers	Free-Range Broilers	Layers	Free-Range Layers	Turkeys	Farmers	Advisors
	(%, 95% CI)								
External biosecurity									
Item E1—animal production on site									
“All-in/all-out” poultry production on site *	68.7 (57.3–80.1)	100	100	100	15.4 (0–38.1)	0	100	76.7 (60.6–92.7)	68 (48.3–87.7)
No backyard on site *	97 (92.8–100)	100	100	100	91.7 (73.3–100)	90 (67.4–100)	100	100	95.5 (86–100)
If other animal production occurs on the site (cattle and pigs); sanitary barriers with poultry (personal, material, etc.)	20.9 (10.9–30.9)	14.3 (0–49.2)	15.4 (0–38.1)	25 (0–63.7)	16.7 (0–41.4)	20 (0–50.2)	38.5 (7.9–69.1)	3.3 (0–10.1)	40.9 (18.6–63.2) ^
Item E2—structure and circulation on site									
Delimitation with a barrier or the closure of a professionally secured area with only necessary vehicles accessing the poultry house (transport vehicles for feed, chicks, poultry or eggs) *	85.1 (76.3–93.8)	100	86.4 (61.9–100)	87.5 (57.9–100)	75 (46.3–100)	70 (35.4–100)	92.3 (75.5–100)	100 ^	72.7 (52.5–92.9)
Wheel dips for the disinfection of the vehicles or pulverization before entering on the site *	95.5 (90.4–100)	100	86.4 (61.9–100)	87.5 (57.9–100)	100	100	100	100	90.9 (77.9–100)
Item E3—personnel, visitors or teams									
Personnel—specific clothes used before entering the house	74.6 (63.9–85.3)	71.4 (26.3–100)	76.9 (50.4–100)	50 (5.3–94.7)	75 (46.3–100)	70 (35.4–100)	84.6 (61.9–100)	90 (78.6–100) ^	63.3 (41.8–85.5)
Personnel—specific shoes used before entering the house *	97.0 (92.8–100)	100	100	100	91.7 (73.3–100)	90 (67.4–100)	100	100	95.5 (86–100)
Personnel—hands washed before entering the house *	52.2 (40–64.5)	85.7 (50.8–100)	46.2 (14.8–77.5)	37.5 (0–80.8)	50 (16.8–83.2)	50 (12.3–87.7)	38.5 (7.9–69.1)	80 (64.8–95.2) ^	27.3 (7.1–47.5)
Personnel—showering before entering the house	10.4 (2.9–18)	71.4 (26.3–100)	0	0	8.3 (0–26.7)	10 (0–32.6)	0	13.3 (0.4–26.2)	13.6 (0–29.2)
Visitors or teams—registration for visitors and teams *	94 (88.2–99.9)	100	100	100	83.3 (58.6–100)	80 (49.8–100)	100	100	90.9 (77.9–100)
Visitors or teams—specific clothes worn before entering the house	85.1 (76.3–93.8)	71.4 (26.3–100)	84.6 (61.9–100)	75 (36.3–100)	91.7 (73.3–100)	90 (67.4–100)	84.6 (61.9–100)	100	72.7 (52.5–92.9)
Visitors or teams—specific shoes worn before entering the house *	97 (92.8–100)	100	100	100	91.7 (73.3–100)	90 (67.4–100)	100	100	95.5 (86–100)
Visitors or teams— hands washed before entering the house *	52.2 (40–64.5)	85.7 (50.8–100)	53.8 (22.5–85.2)	37.5 (0–80.8)	58.3 (25.6–91.1)	40 (3.1–76.9)	38.5 (7.9–69.1)	80 (64.8–95.2) ^	27.3 (7.1–47.5)
Visitors or teams—showering before entering the house	11.9 (4–19.9)	85.7 (50.8–100)	0	0	8.3 (0–26.7)	10 (0–32.6)	0	16.7 (2.5–30.8)	13.6 (0–29.2)
Item E4—the poultry upon arrival									
Flock registration (origin, number of poultry, etc.) *	100	100	100	100	100	100	100	100	100
If the chick deliverer enters in the house, whether specific clothes and shoes are worn *	40.3 (28.2–52.3)	42.9 (0–92.3)	30.8 (1.7–59.8)	37.5 (0–80.8)	41.7 (8.9–74.4)	70 (35.4–100)	30.8 (1.7–59.8)	33.3 (15.4–51.2)	50 (27.3–72.7)
Item E5—the feed and drinking water of the poultry									
Feed storage protection	100	100	100	100	100	100	100	100	100
End-line drinking water analysis performed each year	68.7 (57.3–80.1)	71.4 (26.3–100)	69.2 (40.2–98.3)	75 (36.3–100)	66.7 (35.4–98)	50 (12.3–87.7)	76.9 (50.4–100)	83.3 (69.2–97.5) ^	50 (27.3–72.7) ^
Item E6—biological vector control									
Rodent control (deratting or other measures) *	100	100	100	100	100	100	100	100	100
Wild bird control (protecting the ventilation circuit or other measures) *	100	100	100	100	100	100	100	100	100
No domestic animals on site (pets, dogs or cats)	65.7 (54–77.3)	85.7 (50.8–100)	61.5 (30.9–92.1)	50 (5.3–94.7)	66.7 (35.4–98)	60 (23.1–96.9)	65.1 (30.9–92.1)	90 (78.6–100) ^	45.5 (22.9–68.1)
Item E7—the management of poultry manure									
Manure is stored in a specific isolated area outside of the secured professional area (or if there is no secured area, away from the house)	58.2 (46.1–70.3)	42.9 (0–92.3)	38.5 (7.9–69.1)	62.5 (19.2–100)	66.7 (35.4–98)	60 (23.1–96.9)	76.9 (50.4–100)	66.7 (48.8–84.6)	54.6 (31.9–77.1)
Item E8—the management of dead animals									
Carcasses are removed at least twice a day *	86.6 (78.2–95)	85.7 (50.8–100)	92.3 (75.5–100)	100	75 (46.3–100)	70 (35.4–100)	92.3 (75.5–100)	90 (78.6–100)	81.8 (64.3–99.3)
The presence of a closed and protected rendering tank *	97 (92.8–100)	100	100	100	91.7 (73.3–100)	90 (67.4–100)	100	100	95.5 (86–100)
The rendering tank is located outside of the secured area (or if there is no secured area, away from the house), allowing for the passage of the truck away from the house *	68.7 (57.3–80.1)	85.7 (50.8–100)	76.9 (50.4–100)	62.5 (19.2–100)	58.3 (25.6–91.1)	60 (23.1–96.9)	69.2 (40.2–98.3)	80 (64.8–95.2)	68.2 (47–89.4)
The rendering tanks is cleaned and disinfected after each collection	86.6 (78.2–95)	100	92.3 (75.5–100)	87.5 (57.9–100)	83.3 (58.6–100)	60 (23.1–96.9)	92.3 (75.5–100)	93.3 (83.9–100)	81.8 (64.3–99.3)
Internal biosecurity									
Item I1—structure and circulation in the poultry house									
Concrete surrounds around the house *	47.8 (35.5–60)	71.4 (26.3–100)	38.5 (7.9–69.1)	25 (0–63.7)	50 (16.8–83.2)	60 (23.1–96.9)	33.3 (2.4–64.6)	66.7 (48.8–84.6) ^	36. (14.5–58.2)
Hygiene lock with two separated zones (clean and dirty area) *	67.2 (55.6–78.7)	85.7 (50.8–100)	76.9 (50.4–100)	62.5 (19.2–100)	66.7 (35.4–98)	60 (23.1–96.9)	46.2 (14.8–77.5)	86.7 (73.8–99.6) ^	54.5 (31.9–77.1)
Item I2—the management of the material or litter in the poultry house									
Recognizable separate material only for the poultry house	70.2 (58.9–81.4)	71.4 (26.3–100)	69.2 (40.2–98.3)	75 (36.3–100)	50 (16.8–83.2)	50 (12.3–87.7)	92.3 (75.5–100)	93.3 (83.9–100) ^	50 (27.3–72.7)
The protection of the litter (in a closed shed or other protection, from birds or vermin, etc.) *	67.2 (55.6–78.7)	71.4 (26.3–100)	76.9 (50.4–100)	100	50 (16.8–83.2)	70 (35.4–100)	61.5 (30.9–92.1)	43.3 (24.5–62.2)	86.4 (70.8–100) ^
Item I3—Cleaning and disinfection of the house and material									
The house is cleaned and disinfected between each flock *	100	100	100	100	100	100	100	100	100
The material between each flock (feeders, drinkers, nests, material for the management of eggs, etc.) is cleaned and disinfected *	100	100	100	100	100	100	100	100	100
The drinking water pipeline is cleaned and disinfected between each flock *	95.5 (90.4–100)	100	92.3 (75.5–100)	100	91.7 (73.3–100)	90 (67.4–100)	100	100	90.9 (77.9–100)
The feed silo is cleaned and disinfected between each flock *	74.6 (63.9–85.3)	100	69.2 (40.2–98.3)	50 (5.3–94.7)	75 (46.3–100)	70 (35.4–100)	84.6 (61.9–100)	86.7 (73.8–99.6)	63.6 (41.8–85.5)
There is bacterial auto-control of the cleaning and disinfection of the house between each flock *	16.4 (7.3–25.5)	57.1 (7.7–100)	0	0	33.3 (2–64.6)	30 (0–64.6)	0	20 (4.8–35.2)	13.6 (0–29.2)
There is a sanitary break period > 15 days between each flock *	86.6 (78.2–95)	100	65.1 (30.9–92.1)	75 (36.3–100)	91.7 (73.3–100)	90 (67.4–100)	100	96.7 (89.8–100) ^	72.7 (52.5–92.9)
Item I5—the management of the poultry									
Vaccination protocol of each poultry flock	86.6 (78.2–95)	100	69.2 (40.2–98.3)	62.5 (19.2–100)	91.7 (73.3–100)	90 (67.4–100)	100	96.7 (89.8–100)	77.3 (58.3–96.3)
Daily surveillance with clinical alert criteria (water and feed consummation, mortality and egg production)	100	100	100	100	100	100	100	100	100

Ducks were excluded due to the low number of respondents (*n* = 3 farmers); * fully or partly mandatory BMs according to the Italian legislation; ^ statistically significant difference between farmers and advisors.

## Data Availability

Some of the data presented in this study are available upon request from the corresponding author. The data are not publicly available due to privacy concerns.

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
