# Peer review of "Stakeholders’ Perceptions of Biosecurity Implementation in Italian Poultry Farms"

_animals, 2023, doi:10.3390/ani13203246_

Round 1

Reviewer 1 Report (Previous Reviewer 3)

Interesting paper with much useful and valuable information.

Author Response

Please, see the attached file

Reviewer 2 Report (Previous Reviewer 4)

Dear Authors

Thank you for addressing the recommendations made in the previous submissions.

Regards

Author Response

Please, see the attached file

Reviewer 3 Report (New Reviewer)

Dear authors,

in your text, you presented the results of research related to the perception of different stakeholders (farmers and advisors) on the application of biosecurity measures on poultry farms in Italy. The paper is well written, the English language is appropriate, all parts of the paper smoothly lead to the discussion and conclusions. A shortcoming may be the smaller number of surveyed stakeholders, but they are well distributed by types of poultry and production methods.

Specific comments:

- if not limited by word counting, add a introductory sentence to the abstract

-L102 two commas

-L131 "efficacy of not implemented BMs" - consider to rephrase this part of the sentence (also in L 223-225); 

-L292 dot missing

Author Response

Please, see the attached file

This manuscript is a resubmission of an earlier submission. The following is a list of the peer review reports and author responses from that submission.

Round 1

Reviewer 1 Report

The article provides nice insights about the perceptions of farmers and advisors working for the Italian poultry sector. The authors were careful in addressing the limitations of the study and made use of simple yet sound statistical methods.

General comment: I would include a small paragraph summing up the limitations of this study (e.g. groups size) and address how you dealt with it. Probably connect it with the part where you discuss the EU 

Specific comments:

-Line 23: omit "Indeed". 

Line 23: please rephrase the sentence. The phrase "perceive lower level of biosecurity implementation" might mix up the readership

Line 127: you did not use both simultaneously, please specify and add the justification (for) when using the Fisher's exact test. 

Line 164: difference

Reviewer 2 Report

Dear authors,

thank you for this manuscript. I have read it with great attention and believe it gives some valuable insights. Although I suggest adjustments/additions before it can be accepted for publication.

The paper investigates the implementation of biosecurity measures in different Italien poultry species, including the reason behind the lack of implementation. The results show an average high implementation level, although this might be overestimated by the farmers (compared to the advisors), which again have diverse reasons for not implementing certain aspects. In general breeder production showed the highest implementation level, as well as an overall higher attention for internal biosecurity measures. 

I added some specific comments in the manuscript but have some general thoughts as well.

For me it was not sure if the farmers and advisers rated the same farms. I.e. can you really say the farmers overestimate the implementation or is it possible that the advisors rated other farms with indeed lower BM implementation? Even more clear: did the study include 67 different farms?

Why did you choose for 70% as the cut-off level for high implementation? Please elaborate

How was the information validated? The authors adress the potential bias due to some mandatory measures but did the authors investigated whether the answers corresponded to the actual situation by visiting the farm? 

In general I believe the discussion is written suite defensively towards the farmers. Of course, some aspects of biosecurity might be more difficult to implement on different farms but that does not mean that the risk does not remain. An overview of the "other reasons" for not implementing BM should be added to the supplementary material as well. 

The entire manuscript should be checked by a native speaker.

Especially the result section is sometimes very difficult to understand.

Reviewer 3 Report

This paper discusses the perception of biosecurity implementation in the Italian poultry sector based on a survey of poultry farmers and advisors.  This is timely information given the current avian influenza challenges around the world and the ongoing threats from Salmonella and Campylobacter associated with poultry meat and egg contamination.  The authors use a well-designed survey questionnaire to assess the level of external and internal biosecurity in place on a variety of Italian integrated poultry farms.  It reveals some interesting findings regarding the differences in how farmers and advisors perceive biosecurity implementation, reasons for not implementing certain biosecurity measures, and the level of effectiveness of various biosecurity practices.

The paper presents useful information in how farmers and advisors perceive biosecurity practices which are critical to keeping poultry flocks disease free.  As one might expect due to its importance, the “breeders” group implemented the highest levels of biosecurity.  It is not surprising that free-range layer farms declared the lowest level of biosecurity practices, even though they appeared to understand the importance of biosecurity.

While not indicated, it would be useful to know how farmers were made aware of biosecurity practices.  Did they receive training from advisors or perhaps from integrated companies that they grow birds for?  Is the training ongoing with occasional updates or is biosecurity training a one-time occurrence?  Are biosecurity trainings done in a group setting such as a growing meeting or are trainings done at the individual farmer level?  Similar questions could be asked of advisors. How and by whom are advisors trained and what requirements are met to become biosecurity advisors?  Such information could be useful in assessing how farmers answer a particular question.

Line 77 – How are farmers initially trained on these various biosecurity levels and to what degree and by whom?

Line 140-143 – can you elaborate on why this is?  Is the breakdown related to manure management, dead bird management or something else?  Is it related to their level of training?

Figure 4a – a large percentage of farmers have “not useful” listed as a reason for not implementing some biosecurity practices.  Are these biosecurity practices recommended and not enforced by the integrators or are the integrators nor recommending specific practices and leaving it at the discretion of the farmer?

Line 375 – this difference in perception between farmers and advisors should offer an opportunity to recognize that additional farmer training on biosecurity measures may be in order.  Who should be in charge of this additional training…is this the advisors or the integrators?

This is a useful paper with valuable information related to the perception of farmers and advisors concerning biosecurity in the Italian poultry industry.  It indicates the awareness and importance of biosecurity but also recognizes where gaps exist.  These gaps indicate the need for additional outreach and training to enhance biosecurity measures.

Reviewer 4 Report

Dear Authors

The manuscript  Farmers’ and Advisors’ Perception of Biosecurity Implementation in the Italian Poultry Sector focuses on the application of interviews to know the degree of implementation of biosecurity measures (BM) in poultry producers in Italy, as well as the reasons for implementing them or not. Among the main results, the authors mention the degree of implementation of BM and the differences in the perception of the advantages and/or reasons for the implementation of BM, between farmers and advisors.

The article has several important points that should be improved in order to be accepted for publication:
- The authors do not mention the statistical criteria used to choose the number of stakeholders to interview.
- There is also no mention of the criteria considered for selecting the farms to be included in the study, the professional level of the advisors or their specialty.
- Similarly, there is no explanation of the region in which the study was carried out. The title indicates the Italian poultry sector, however, due to the number of people interviewed, I do not believe that it is representative of the Italian poultry sector.
- With the data shown in this article, it is not possible to conclude the degree of understanding and perception of the implementation of the BM, nor to identify the reasons why they are not implemented. To make a conclusion as described by the authors, a questionnaire with a representative number of poultry producers needs to be carried out.
- Another type of figure should be chosen for a better understanding of the data shown in Figures 2, 4 and 5.
Kind Regards

Round 2

Reviewer 2 Report

Thank you for answering all my comments.

I believe supplementary table 3 is missing however.

Reviewer 4 Report

Dear Authors

I still have the same doubts:

- There is also no mention of the criteria considered for selecting the farms to be included in the study, the professional level of the advisors or their specialty.
- Similarly, there is no explanation of the region in which the study was carried out. The title indicates the Italian poultry sector, however, due to the number of people interviewed, I do not believe that it is representative of the Italian poultry sector.
- With the data shown in this article, it is not possible to conclude the degree of understanding and perception of the implementation of the BM, nor to identify the reasons why they are not implemented. To make a conclusion as described by the authors, a questionnaire with a representative number of poultry producers needs to be carried out.

Best regards